# Variable Step-Size Hybrid Filtered-x Affine Projection Generalized Correntropy Algorithm for Active Noise Control

**DOI:** 10.3390/s25061881

**Published:** 2025-03-18

**Authors:** Zhaoqing Mu, Ying Gao, Xinyu Guo, Shifeng Ou

**Affiliations:** 1School of Physics and Electronic Information, Yantai University, Yantai 264005, China; muzhaoqing@s.ytu.edu.cn (Z.M.); claragaoying@126.com (Y.G.); 2Key Laboratory of Geophysical Exploration Equipment, Ministry of Education, College of Instrumentation and Electrical Engineering, Jilin University, Changchun 130000, China; xyguo22@mails.jlu.edu.cn

**Keywords:** industrial noise reduction, ANC, impulse noise, generalized maximum correntropy, variable step size

## Abstract

Active Noise Control (ANC) is frequently utilized to minimize noise in industrial environments. However, the powerful pulses in industrial noise pose challenges to its application. Consequently, ANC systems necessitate a high-performance algorithm as a core component. In this process, the variable step-size strategy is the main approach for enhancing the ANC algorithm’s performance but ensuring robustness while improving performance remains a challenge. To address this problem, we propose a new ANC algorithm with a variable step size. This algorithm is derived from the Affine Projection Generalized Maximum Correntropy (APGMC) method, featuring a hybrid step-size and a new step-size approach achieved by modifying the mean square deviation (MSD). To showcase the practical effectiveness of the proposed algorithm, noisy audio from a real construction site was used for noise reduction control. Results show that the proposed algorithm effectively manages noise across frequency bands, with an improvement of approximately 16% to 19.2% compared to existing similar algorithms.

## 1. Introduction

ANC is a widely used technique for noise suppression. It works by superimposing sound waves, performing inverse superposition in real-world settings to reduce or eliminate noise. The ANC system mainly consists of the following important components: a reference sensor for picking up the sound signals, a controller for adjusting the parameters of the filter, a loudspeaker for generating the inverse noise signals, and an error sensor for detecting the residual signals [1]. The ANC technique is used in many scenarios, such as personal audio devices [2], vehicle and aviation noise reduction [3,4], and industrial noise reduction [5,6].

Adaptive filtering (AF) algorithms are often used in signal processing tasks like system identification, Acoustic Echo Cancellation (AEC), and ANC [7,8,9]. AF applied to ANC often uses variable step size to improve performance [10,11]. However, real industrial noise often has impulsive characteristics, which can reduce the effectiveness of variable step-size methods in this application scenario. Therefore, designing a variable step-size ANC algorithm that takes robustness into account is the problem to be solved.

Popular AF algorithms for impulse noise are generally grouped into three main categories: low-order moment algorithms [12,13,14], nonlinear transformation algorithms [15,16,17], and information-theoretic learning algorithms [18,19,20]. The mixed-norm algorithm [21,22] is one of the more prominent and advanced of the low-order moment algorithms. Specifically, ref. [22] introduced the hybrid step-size concept in the mixed-norm algorithm, enhancing its effectiveness in dealing with impulsive noise. Subsequently, ref. [23] introduced innovations to the variable step size in [22], resulting in additional performance improvements. Naturally, the algorithm proposed in [23] has become one of the most advanced approaches in impulsive noise processing. However, the existence of hybrid norms dictates that only a compromise between algorithmic performance and robustness can be made, which is a drawback of the current class of algorithms. The current nonlinear transformation algorithms commonly used are the data selection algorithm [24] and trigonometric function algorithms [25], which have excellent performance, but need to set the a priori parameters, and cannot be adjusted in real time when the signal changes. Representative information-theoretic learning algorithms such as APGMC [26] and M-estimation-based minimum error entropy (MMEE) [27] are more robust and do not need to set a priori parameters, but also inevitably increase the computational complexity. In light of these considerations, optimizing algorithm performance while reducing computational complexity presents a key area for future research in information-theoretic learning algorithms.

Commonly used variable step-size methods are minimized MSD design methods [23,28,29,30], modified MSD design methods [31,32], and other design methods such as trigonometric functions [33] and switching strategy [34,35]. Although the moving average method can help such algorithms effectively avoid outliers and improve the smoothness of the step size, it still results in some performance loss for such algorithms when handling real impulsive industrial noise. Additionally, the variable step-size algorithm that relies on the trigonometric function [36] still faces the challenge of setting the a priori parameter, which is not an optimal choice.

Based on the summary and considerations of the aforementioned robust algorithms and adaptive variable step-size methods, designing a novel, robust, and efficient variable step-size ANC algorithm remains a challenging task. This requires not only ensuring the algorithm’s robustness but also preserving the stability of the variable step-size scheme. Our goal is to address both aspects simultaneously, rather than making a trade-off between them. In particular, variable step sizes are designed to minimize performance loss due to outliers in the pulse signal. The contributions of this paper are primarily twofold: First, based on the boundary analysis of the step size in the APGMC algorithm, a hybrid step-size scheme suitable for this algorithm is proposed, leading to a variant algorithm with improved performance. At the same time, this variant algorithm also provides an algorithmic foundation for developing variable step-size strategies. Second, the introduction and implementation of the variable step-size scheme are presented. A thorough analysis of the MSD for the variant algorithm is carried out, and from this, a controllable error term is incorporated into the design, ensuring both efficiency and stability in each iteration. Additionally, real-world noise audio collected from a construction site is utilized to evaluate performance, thereby highlighting the effectiveness and value of the contributions made in this paper.

From a scientific perspective, the reduction in industrial noise through Active Noise Control (ANC) relies on robust ANC algorithms as a key enabler. Although there are some effective techniques and algorithms available, from the physical sounding process, construction generates impulsive noise with huge power [37], which is already a challenge for the real-time noise control. From a societal perspective, industrial noise not only negatively affects the health of employees [38,39] but also impacts the quality of life of surrounding communities [40,41] and the sustainable development of the social economy [42]. Therefore, controlling and eliminating industrial noise is of significant importance for improving human life quality and promoting social and economic development.

The structure of this paper is as follows: Section 2 provides an overview of the ANC model and the APGMC algorithm. Section 3 sequentially presents the variant algorithm, the variable step-size algorithm, and a detailed stability performance analysis. Section 4 presents the results, which include an analysis of real industrial noise and an assessment of the proposed algorithm’s performance. Section 5 wraps up the paper and offers recommendations for future research.

## 2. Algorithm and System Review

The ANC algorithm refers to the application of adaptive filtering algorithms in the ANC model. Therefore, we have reviewed the ANC system model as well as the APGMC algorithm to be improved. Figure 1 shows the schematic of the APGMC algorithm applied to the ANC system [43], where **P**(*z*) and **S**(*z*) are the transfer functions of the primary and secondary paths, **W**(*z*) and **d**(*n*) are the ANC controller and the primary noise to be cancelled, respectively, and $\tilde{S}(z)$ estimates **S**(*z*), and the input reference signal **U**(*n*) of FxAPGMC needs to be filtered by $\tilde{S}(z)$ and becomes $U^{\prime }(n)$. **y**(*n*) and **e**(*n*) denote the filter response and error signal, respectively, and **v**(*n*) represents background noise.

As the response, **y**(*n*) is expressed as(1)$$y(n)=U^{T}(n)w(n)$$
where $U(n)=\left[u(n),u(n-1),\;\ldots ,u(n-M+1)\right]$, $u(n)={\left[u(n),u(n-1),\;\ldots ,u(n-L+1)\right]}^{T}$, *M* represents the projection order of **U**(*n*), while *L* is the length of **w**(*n*).

The **e**(*n*) is calculated using the following expression:(2)$$e(n)=d(n)-y^{\prime }(n)$$
where d(n)=U(n)*p(n)+v(n) and $y^{\prime }(n)=y(n)*s(n)$, while **p**(*n*) and **s**(*n*) represent the impulse responses of **P**(*z*) and **S**(*z*), and $U^{\prime }(n)=U(n)*s(n)$ is **U**(*n*) filtered by a secondary estimated path. The symbol * indicates the discrete convolution operator.

According to GMCC [18], the generalized correntropic loss function is(3)$$J_{GC}(X-Y)=\gamma -V_{GC}(X-Y)$$
where *X* and *Y* represent two different random variables, γ is a normalization constant, $V_{GC}(X-Y)=\gamma E[\exp(-\phi {\left|X-Y\right|}^{\alpha })]$, and ϕ is a parameter that controls shape and scale, and we can estimate $V_{GC}(X-Y)$ through the sample pairs of random variables (*X*, *Y*), and then Equation (3) becomes(4)$${\hat{J}}_{GC}^{N}(X-Y)=\gamma -\frac{1}{N}\sum_{j=1}^{N}\gamma \exp(-\phi {\left|x_{i}-y_{i}\right|}^{\alpha })$$

while γ is a constant, minimizing ${\hat{J}}_{GC}^{N}(X-Y)$ is equivalent to minimizing(5)$$1-\frac{1}{N}\sum_{j=1}^{N}\exp(-\phi {\left|x_{j}-y_{j}\right|}^{\alpha })$$

Combining the affine projection method and GMCC according to [26], we can obtain an optimization model for the FxAPGMC as(6)$$\underset{w(n+1)}{\min}M-\sum_{j=0}^{M-1}\exp(-\phi {\left|e_{p}(n-j)\right|}^{\alpha })\;\mathrm{subject}\;\mathrm{to}\;{\left\|w(n+1)-w(n)\right\|}_{2}^{2}\leq \mu^{2}$$
where $e_{p}(n-j)$ is a posterior error and $e_{p}(n-j)$ can be approximately estimated by e(n−j), with μ being the parameter used to ensure the stability of the vector of weighting factors, and it is commonly referred to as the step size. We adopt the Lagrange multiplier method to convert Equation (6) to unconstrained optimization, and we calculate the gradient of **w**(*n* + 1) to obtain(7)$$\nabla J(w(n+1))=-\alpha \phi \sum_{j=0}^{M-1}\left(b(n-j)\mathrm{sgn}(e_{p}(n-j)U^{\prime }(n-j)\right)+2\theta (w(n+1)-w(n))$$
where θ is a Lagrange multiplier, $b(n-j)=\exp(-\phi {\left|e(n-j)\right|}^{\alpha }){\left|e(n-j)\right|}^{\alpha -1}$, let Equation (7) be zero and combine Equation (6), and the formula for updating the weight of FxAPGMC can be obtained as(8)$$w(n+1)=w(n)+\mu \frac{U^{\prime }(n)B(n)\mathrm{sgn}(e(n))}{{\left\|U^{\prime }(n)B(n)\mathrm{sgn}(e(n))\right\|}_{2}}$$
where **B**(*n*) = *diag* [*b*(*n*), *b*(*n* − 1), …, *b*(*n* − *M* + 1)], and sgn(·) is the sign function.

The FxAPGMC algorithm, although effective in addressing impulse noise issues, still suffers from high computational complexity and slow convergence speed. In practical construction scenarios, noise is often complex and variable. During construction, noise with impulsive characteristics is particularly common. For instance, in subway construction, pile drivers are typically used for foundation reinforcement or to support wall construction. The impulsive noise generated during this process is unavoidable and often accompanied by high power, resulting in significant noise pollution to the surrounding environment. We aim to propose an ANC algorithm that can more effectively handle such types of noise. Therefore, this paper introduces a hybrid step-size variant of the algorithm, presenting an improved step-size scheme with better performance.

Although the FxAPGMC algorithm can solve the impulse noise problem, it suffers from high computational complexity and slow convergence speed in dealing with real industrial noise. In practical applications, we aim for the proposed ANC algorithm to deliver improved noise reduction. Therefore, this paper carries out a hybrid step-size variant of this algorithm and proposes a variable step-size scheme with better performance.

## 3. The Proposed Algorithm

### 3.1. Hybrid Scheme

The hybrid step-size can give the original algorithm a performance boost. For example, H-APA [22] proposed combining the step sizes of APSA [29] and RMNA [21], which improves the robustness of the original algorithm under impulse noise as well as the convergence speed. Inspired by this, this paper subjects the FxAPGMC algorithm to hybrid step-size processing. The gainful effects that the hybrid scheme can bring should include the following: improving the convergence speed of the original algorithm, reducing the computational complexity, and ensuring the robustness of the algorithm, as well as the need to satisfy the multidimensional properties. Therefore, the step size of MFxAPSA is used to be hybridized with the FxAPGMC algorithm.

In the above, consider $\mu /{\left\|U^{\prime }(n)B(n)\mathrm{sgn}(e(n))\right\|}_{2}$ in Equation (8) as a step size of the FxAPGMC algorithm to be modified; according to [29], the weight formula of MFxAPSA can be expressed as(9)$$w(n+1)=w(n)+\mu^{\prime }\frac{U^{\prime }(n)\mathrm{sgn}(e(n))}{{\left\|U^{\prime }(n)\mathrm{sgn}(e(n))\right\|}_{2}}$$
where $\mu^{\prime }/{\left\|U^{\prime }(n)\mathrm{sgn}(e(n))\right\|}_{2}$ in Equation (9) can be considered as a special step size, and the **B**(*n*) is closely related to **e**(*n*) in Equation (8). We can therefore consider an inequality relation between $\mu /{\left\|U^{\prime }(n)B(n)\mathrm{sgn}(e(n))\right\|}_{2}$ and $\mu^{\prime }/{\left\|U^{\prime }(n)\mathrm{sgn}(e(n))\right\|}_{2}$ [22], and try to utilize the unequal relationship to replace the above step size, so that we find a $\mu^{\prime }$ at the beginning of the iteration, such that(10)$$\frac{\mu }{{\left\|U^{\prime }(n)B(n)\mathrm{sgn}(e(n))\right\|}_{2}}\leq \frac{\mu^{\prime }}{{\left\|U^{\prime }(n)\mathrm{sgn}(e(n))\right\|}_{2}}\leq \frac{\mu_{\max}}{{\left\|U^{\prime }(n)B(n)\mathrm{sgn}(e(n))\right\|}_{2}}$$
where $\mu^{\prime }\geq (\mu {\left\|U^{\prime }(n)\mathrm{sgn}(e(n))\right\|}_{2})/({\left\|U^{\prime }(n)B(n)\mathrm{sgn}(e(n))\right\|}_{2})$ can serve to achieve fast convergence at the beginning of the iteration. When α and ϕ in **B**(*n*) are concrete parameters [26], it can be determined that **B**(*n*) is bounded, and therefore, the $\mu^{\prime }$ can be found such that $\mu^{\prime }\leq \mu_{\max}{\left\|U^{\prime }(n)\mathrm{sgn}(e(n))\right\|}_{2}/{\left\|U^{\prime }(n)B(n)\mathrm{sgn}(e(n))\right\|}_{2}$.

To enable the FxAPGMC algorithm’s robustness and fast convergence, we replace the step size of the FxAPGMC algorithm with the special step size of MFxAPSA. Then, the weight equation of the variant algorithm is obtained:(11)$$w(n+1)=w(n)+\mu \frac{U^{\prime }(n)B(n)\mathrm{sgn}(e(n))}{{\left\|U^{\prime }(n)\mathrm{sgn}(e(n))\right\|}_{2}}$$

In addition, we need to add a regularization factor σ to the denominator of Equation (11) to avoid zero, and the weight equation can be written as(12)$$w(n+1)=w(n)+\mu \frac{U^{\prime }(n)B(n)\mathrm{sgn}(e(n))}{{\left\|U^{\prime }(n)\mathrm{sgn}(e(n))\right\|}_{2}+\sigma }$$
where σ is a positive number close to zero.

This variant scheme enhances the effectiveness of the FxAPGMC algorithm, particularly in computational complexity and convergence speed (this part is demonstrated in the impulse noise simulation). Furthermore, it provides the basis for the development of a new variable step-size algorithm.

### 3.2. The Proposed VSS-H-FxAPGMC

In this section, we first define $w^{o}$ as the optimal weight and $\tilde{w}(n+1)≜w^{o}-w(n+1)$ as the weight error. Substituting this into Equation (11), we can obtain(13)$$\tilde{w}(n+1)=\tilde{w}(n)-\mu (n)\frac{U^{\prime }(n)B(n)\mathrm{sgn}(e(n))}{{\left\|U^{\prime }(n)\mathrm{sgn}(e(n))\right\|}_{2}}$$

Next, by taking the expectation of both sides of the square in Equation (13), the MSD update recursion is derived:(14)$$\begin{array}{c} E\left[{\left\|\tilde{w}(n+1)\right\|}^{2}\right]=E\left[{\left\|\tilde{w}(n)\right\|}^{2}\right]-2E[\mu (n)]E\left[\frac{{\mathrm{sgn}}^{T}(e(n))B(n)U^{\prime T}(n)\tilde{w}(n)}{{\left\|U^{\prime }(n)\mathrm{sgn}(e(n))\right\|}_{2}}\right]\\ +E[\mu^{2}(n)]E\left[\frac{{\left\|U^{\prime }(n)B(n)\mathrm{sgn}(e(n))\right\|}_{2}^{2}}{{\left\|U^{\prime }(n)\mathrm{sgn}(e(n))\right\|}_{2}^{2}}\right] \end{array}$$
where $E\left[{\left\|\tilde{w}(n)\right\|}^{2}\right]$ can be considered as the distance between $w^{o}$ and **w**(*n*). When the filter converges, we can obtain(15)$$\underset{n\to \infty }{\lim}E\left[{\left\|\tilde{w}(n+1)\right\|}^{2}\right]=\underset{n\to \infty }{\lim}E\left[{\left\|\tilde{w}(n)\right\|}^{2}\right]$$

For the purpose of simplifying the derivation, we assume the following: μ(n) is statistically independent of all other variables in the proposed algorithm, **w**(*n*) is independent of $U^{\prime }(n)$, and $\tilde{w}(n)$ is independent of **e**(*n*) [31]. The combination of the form of the update recursion of MSD in Equation (14) with the ideal state of Equation (15) can be given by(16)$$E[\mu (\infty )]=2E\left[\frac{{\left\|U^{\prime }(\infty )\mathrm{sgn}(e(\infty ))\right\|}_{2}{\mathrm{sgn}}^{T}(e(\infty ))B(\infty )U^{\prime T}(\infty )\tilde{w}(\infty )}{{\left\|U^{\prime }(\infty )B(\infty )\mathrm{sgn}(e(\infty ))\right\|}_{2}^{2}}\right]$$

However, μ(∞) in Equation (16) cannot be a wise choice at any time *n* because it denotes the steady-state step size. The control term should be a positive number positively correlated with **e**(*n*) to ensure rapid convergence in the initial stages of the iteration. A similar method for modifying MSD equations using control terms exists in [31]. Therefore, we designed a control term to improve Equation (16) and the control term can be expressed as(17)$$r(n)=\eta r(n-1)+(1-\eta ){\left\|e(n)\right\|}_{2}^{2}$$
where η denotes the smoothing parameter with a range from 0 to 1 and controls the weight of error signal **e**(*n*). Then, introducing Equation (17) to Equation (16), the step-size μ(n) followed as(18)$$\mu (n)=r(n)E\left[\frac{{\left\|U^{\prime }(n)\mathrm{sgn}(e(n))\right\|}_{2}{\mathrm{sgn}}^{T}(e(n))B(n)U^{\prime T}(n)\tilde{w}(n)}{{\left\|U^{\prime }(n)B(n)\mathrm{sgn}(e(n))\right\|}_{2}^{2}}\right]$$

By combining Equations (1) and (2), $U^{\prime T}(n)$ in Equation (18) can be expressed as follows: (19)$$U^{\prime T}(n)\tilde{w}(n)=e(n)-v(n)$$

and the update equation of μ(n) can be obtained:(20)$$\mu (n)=r(n)E\left[\frac{{\left\|U^{\prime }(n)\mathrm{sgn}(e(n))\right\|}_{2}{\mathrm{sgn}}^{T}(e(n))B(n)(e(n)-v(n))}{{\left\|U^{\prime }(n)B(n)\mathrm{sgn}(e(n))\right\|}_{2}^{2}}\right]$$

However, mathematical expectation is required in Equation (20), and determining μ(n) directly is challenging. Hence, the moving average method is applied to handle the expectation term. In this process, we set a $\mu^{*}(n)$ as the final variable step-size scheme, and the connection between $\mu^{*}(n)$ and the aforementioned μ(n) can be understood as follows: $\mu^{*}(n)$ adopts an approximate form $\mu_{a}(n)$ of μ(n) as its component [28]. This method for $\mu^{*}(n)$ can be expressed as(21)$$\mu^{*}(n)=\left\{\begin{array}{l} \lambda \mu^{*}(n-1)+(1-\lambda )\min(\mu_{a}(n),\mu^{*}(n-1)),\\ \;\;\;\;\;\;\;\;\;\;\;\;\mathrm{if}\;0<\mu_{a}(n)\\ \mu^{*}(n-1),\;\;\;\;\;\;\;\;\mathrm{else} \end{array}\right.$$
where λ (with 0 ≤ λ < 1) represents the smoothing factor. The use of the min function is to make $\mu^{*}(n)$ tend to decrease over the course of each iteration, and the $\mu_{a}(n)$ of μ(n) after ignoring mathematical expectation can be obtained:(22)$$\mu_{a}(n)=r(n)\frac{{\left\|U^{\prime }(n)\mathrm{sgn}(e(n))\right\|}_{2}{\mathrm{sgn}}^{T}(e(n))B(n)(e(n)-v(n))}{{\left\|U^{\prime }(n)B(n)\mathrm{sgn}(e(n))\right\|}_{2}^{2}}$$

Since the moving average method used in Equation (21) is only applicable to smooth environments, and real industrial noise reduction scenarios are often accompanied by suddenness and uncertainty, the proposed method is unable to reset the step size in non-smooth environments, which in turn prevents the algorithm from achieving fast convergence in non-smooth environments. To overcome the weakness of the proposed algorithm in the sudden system change environment, the reset method is used to preserve the algorithm’s effectiveness in the event of a sudden system change, as shown in Algorithm 1.
**Algorithm 1.** Step-size reset method of VSS-H-FxAPGMC algorithm$Initialization\;values:T_{p},T_{d},\varepsilon$$\begin{array}{l} \mathrm{For}\;\mathrm{each}\;n\\ \;\mathrm{if}\;\mathrm{mod}(n,T_{p})=0\\ {\;\;\mathrm{ctrl}}_{\mathrm{new}}=\frac{Q^{T}MQ}{T_{p}-T_{d}}\\ \;\mathrm{end}\\ {\;\mathrm{if}\;(\mathrm{ctrl}}_{\mathrm{new}}-{\mathrm{ctrl}}_{\mathrm{old}})/\mu^{*}(n-1)>\xi \\ \;\;\mu^{*}(n)=\mu^{*}(0)\\ {\;\mathrm{else}\;\mathrm{if}\;\mathrm{ctrl}}_{\mathrm{new}}>{\mathrm{ctrl}}_{\mathrm{old}}\;\\ \;\;\mathrm{Not}\;\mathrm{update}\;\mathrm{in}\;\mathrm{the}\;\mathrm{current}\;\mathrm{iteration}.\\ \;\mathrm{else}\\ \;\;\mathrm{Update}\;\mathrm{the}\;\mathrm{formula}\;\\ \;\mathrm{end}\\ {\mathrm{ctrl}}_{\mathrm{old}}={\mathrm{ctrl}}_{\mathrm{new}}\\ \mathrm{end} \end{array}$

$T_{p}$ and $T_{d}$ are two positive integers ($T_{p}>T_{d}$), and $Q=\;\mathrm{sort}\;[(\left|e(n)\right|)/({\left|\left|u(n)\right|\right|}_{2}$$+\varepsilon ),\;\ldots ,\;(\left|e(n-T_{p}+1)\right|)/({\left\|u(n-T_{p}+1)\right\|}_{2}+\varepsilon ){]}^{T}$, where “sort” refers to the operator for arranging in ascending order, **M** is a diagonal matrix with the first $T_{p}-T_{d}$ elements set to 1, and ξ is a threshold.

After solving the problem of sudden system changes, inserting Equation (22) into Equation (12), we can obtain the weight equation of VSS-H-FxAPGMC as(23)$$w(n+1)=w(n)+\mu^{*}(n)\frac{U^{\prime }(n)B(n)\mathrm{sgn}(e(n))}{{\left\|U^{\prime }(n)\mathrm{sgn}(e(n))\right\|}_{2}+\sigma }$$

### 3.3. Stability Analysis

The algorithm applied to noise reduction in real scenarios should be stable and effective. Therefore, it is essential to analyze the stability of the proposed algorithm. We determine the range of μ(*n*), and the update recursion of MSD is shown in Equation (14), and this recursion should be followed:(24)$$E\left[{\left\|\tilde{w}(n+1)\right\|}^{2}\right]-E\left[{\left\|\tilde{w}(n)\right\|}^{2}\right]\leq 0$$

and μ(*n*) should satisfy the following:(25)$$\mu (n)\leq 2E\left[\frac{{\left\|U^{\prime }(n)\mathrm{sgn}(e(n))\right\|}_{2}{\mathrm{sgn}}^{T}(e(n))B(n)U^{\prime T}(n)\tilde{w}(n)}{{\left\|U^{\prime }(n)B(n)\mathrm{sgn}(e(n))\right\|}_{2}^{2}}\right]$$

To better optimize the step-size range, it is assumed that **v**(*n*) is i.i.d. and independent of $U^{\prime }(n)$. Then, let $f(n)={\mathrm{sgn}}^{T}(e(n))B(n)U^{\prime T}(n)\tilde{w}(n)$ in Equation (25). With the above assumptions, we can initially obtain(26)$$f(n)={\mathrm{sgn}}^{T}(e(n))B(n)e(n)$$

concretize operations between vectors and matrices(27)$$f(n)=\sum_{i=n-M+1}^{n}\mathrm{sgn}(e(i))\exp(-\phi {\left|e(i)\right|}^{\alpha }){\left|e(i)\right|}^{\alpha -1}e(i)=\sum_{i=n-M+1}^{n}\exp(-\phi {\left|e(i)\right|}^{\alpha }){\left|e(i)\right|}^{\alpha }$$

It is straightforward to demonstrate(28)$$0\leq f(n)\leq \frac{M}{\phi }\exp(-1)$$

The range of μ(*n*) after optimization can be expressed as(29)$$\mu (n)\leq \frac{2M}{\phi }\exp(-1)E\left[\frac{{\left\|U^{\prime }(n)\mathrm{sgn}(e(n))\right\|}_{2}}{{\left\|U^{\prime }(n)B(n)\mathrm{sgn}(e(n))\right\|}_{2}^{2}}\right]$$

A thorough analysis of the POD is necessary for gaining a better understanding of the mean square stability of the proposed algorithm [18]. Divergence here refers to the adaptive algorithm implementation of $\underset{n\to \infty }{\lim}{\left\|\tilde{w}(n)\right\|}^{2}=\infty$.

Let us consider the scalar filtering case [18], in which $d(n)=w^{o}U^{\prime }(n)$, where $w^{o}$ and $U^{\prime }(n)$ are both scalars, *B*(*n*) is denoted as a first-order matrix. For simplicity, it is assumed that the noise *v*(*n*) is zero.

Equation (13) can be rewritten as(30)$$\begin{array}{l} \tilde{w}(n+1)=\tilde{w}(n)-\mu (n)\frac{U^{\prime }(n)B(n)\mathrm{sgn}(e(n))}{{\left\|U^{\prime }(n)\mathrm{sgn}(e(n))\right\|}_{2}}\\ =\tilde{w}(n)-\mu (n)\frac{U^{\prime }(n)\exp(-\phi {\left|e(n)\right|}^{\alpha }){\left|e(n)\right|}^{\alpha -1}\mathrm{sgn}(e(n))}{{\left\|U^{\prime }(n)\mathrm{sgn}(e(n))\right\|}_{2}}\\ =\left[1-\mu (n)\frac{\exp(-\phi {\left|e(n)\right|}^{\alpha }){\left|e(n)\right|}^{\alpha -2}U^{\prime 2}(n)}{{\left\|U^{\prime }(n)\mathrm{sgn}(e(n))\right\|}_{2}}\right]\tilde{w}(n)\\ =\left[1-\mu (n)\frac{\exp(-\phi {\left|e(n)\right|}^{\alpha }){\left|e(n)\right|}^{\alpha }{\left|\tilde{w}(n)\right|}^{-2}}{{\left\|U^{\prime }(n)\mathrm{sgn}(e(n))\right\|}_{2}}\right]\tilde{w}(n) \end{array}$$
where $\exp(-\phi {\left|e(n)\right|}^{\alpha }){\left|e(n)\right|}^{\alpha }$ which ranges from 0 to $\frac{1}{\phi }\exp(-1)$ has been mentioned in Equation (28).

So that if ${\left|\tilde{w}(n)\right|}^{2}\geq \frac{\mu (n)\exp(-1)}{2\phi {\left\|U^{\prime }(n)\mathrm{sgn}(e(n))\right\|}_{2}}$, we have(31)$$0\leq \mu (n)\frac{\exp(-\phi {\left|e(n)\right|}^{\alpha }){\left|e(n)\right|}^{\alpha }{\left|\tilde{w}(n)\right|}^{-2}}{{\left\|U^{\prime }(n)\mathrm{sgn}(e(n))\right\|}_{2}}\leq 2$$

In this case, taking the mean square for Equation (30), we can obtain(32)$${\left|\tilde{w}(n+1)\right|}^{2}={\left[1-\mu (n)\frac{\exp(-\phi {\left|e(n)\right|}^{\alpha }){\left|e(n)\right|}^{\alpha }{\left|\tilde{w}(n)\right|}^{-2}}{{\left\|U^{\prime }(n)\mathrm{sgn}(e(n))\right\|}_{2}}\right]}^{2}{\left|\tilde{w}(n)\right|}^{2}\leq {\left|\tilde{w}(n)\right|}^{2}$$

Therefore, if the limit of ${\left|\tilde{w}(n)\right|}^{2}$ exists ($\underset{n\to \infty }{\lim}{\left\|\tilde{w}(n)\right\|}^{2}<\infty$), this would represent that the proposed algorithm will never diverge under ideal circumstances.

The POD of the proposed algorithm will become complicated when **w**(*n*) returns to the vector and noise **v**(*n*) is present. However, experimental data show that the proposed algorithm still guarantees stable convergence in real industrial noise that is complex and variable with outliers.

According to [26], the comprehensive performance of the FxAPGMC algorithm performs optimally when α = 2 and ϕ = 0.01, and the setting of this parameter is unrelated to the step-size design, so that the hybrid step-size does not conflict with the values of α and ϕ. α = 2 and ϕ = 0.01 are also applicable to the proposed VSS-H-FxAPGMC algorithm. Algorithm 2 summarizes the proposed VSS-H-FxAPGMC algorithm.
**Algorithm 2.** VSS-H-FxAPGMC AlgorithmInitialization values:w(0)=0, r(0), and μ(0)$\begin{array}{l} Parameters\;setting:L_{w}=32,\;\alpha =2,\;\phi =0.01\\ \mathrm{For}\;\mathrm{each}\;\mathrm{iteration}\;n\\ \;e(n)=d(n)-U^{\prime T}(n)w(n)\\ \;b(n)=\exp(-\phi {\left|e(n)\right|}^{\alpha }){\left|e(n)\right|}^{\alpha -1}\\ \;B(n)=diag(b(n),b(n-1),\;\ldots ,b(n-M+1))\\ \;r(n)=\eta r(n-1)+(1-\eta ){\left\|e(n)\right\|}_{2}^{2}\\ \;\mu_{a}(n)=r(n)\frac{{\left\|U^{\prime }(n)\mathrm{sgn}(e(n))\right\|}_{2}{\mathrm{sgn}}^{T}(e(n))B(n)(e(n)-v(n))}{{\left\|U^{\prime }(n)B(n)\mathrm{sgn}(e(n))\right\|}_{2}^{2}}\\ \;\mathrm{if}\;\mu_{a}(n)>0\\ \;\;\mu (n)=\lambda \mu (n-1)+(1-\lambda )\min(\mu_{\alpha }(n),\mu (n-1))\\ \;\mathrm{else}\\ \;\;\;\mu (n)=\mu (n-1)\\ \;\mathrm{end}\\ \;w(n+1)=w(n)+\mu (n)\frac{U^{\prime }(n)B(n)\mathrm{sgn}(e(n))}{{\left\|U^{\prime }(n)\mathrm{sgn}(e(n))\right\|}_{2}+\sigma }\\ \mathrm{end} \end{array}$

It is important to describe the general methodology of the experiment, so here we describe the general flow of the derivation of the proposed algorithm, as illustrated in Figure 2.

Figure 2 gives a concise summary of the process and methodology of the proposed algorithm, which was developed in three stages. The first stage involves implementing the original APGMC algorithm in the ANC system, which requires filtered-x processing. The second stage introduces the concept of hybrid step size, including the theoretical foundation for this implementation and the criteria for selecting the step size. The third stage emphasizes designing and implementing a variable step-size strategy, resulting in a final algorithm tailored for real-world industrial noise.

## 4. Experimental Results

In this section, the simulated impulse noise and the performance of the proposed VSS-H-FxAPGMC algorithm is evaluated using real industrial noise. The comparison algorithms include two representative classes of algorithms: the classical ANC robust algorithms VSS-MFxAPSA and VSS-NFxAPSA [29], and the current state-of-the-art robust algorithms VSS-H-APA and MVSS-H-APA after filtered-x in ANC [23]. Among them, VSS-MFxAPSA and VSS-NFxAPSA are classical algorithms widely used in impulse noise processing, hence being necessary as a comparison. VSS-H-APA and MVSS-H-APA are novel algorithms, both of which use a hybrid step-size approach. Among the variable step-size schemes, VSS-H-APA uses the MSD minimization principle and MVSS-H-APA uses a modified MSD method and hence has greater comparative value.

Prior to the comparison, the effectiveness of the hybrid variant algorithms is first validated against the similar algorithms which include MFxAPSA, FxAPGMC and APRMN [21], H-AP [22], and EPAA [44] after filtered-x in ANC.

During the simulation, we set the following conditions to advance the experiment. We assume that **S**(*z*) and $\tilde{S}(z)$ are consistent. The length of **w**(*n*) is designed as *L* = 32, the paths are represented as finite impulse responses (FIRs) with primary and secondary path lengths of 32 and 12, respectively. The averaged noise reduction (ANR), used as the performance evaluation criterion, can be defined as(33)$$\mathrm{ANR}(n)=20\log_{10}\left(\frac{A_{e}(n)}{A_{d}(n)}\right)$$
where $A_{e}(n)=\kappa A_{e}(n-1)+(1-\kappa )\left|e(n)\right|$ and $A_{d}(n)=\kappa A_{d}(n-1)+(1-\kappa )\left|d(n)\right|$ with $A_{e}(0)=A_{d}(0)=0$ and κ = 0.999 can be considered as a forgetting factor. The performance of the simulated algorithms will be compared based on convergence speed and steady-state ANR. Additionally, all simulation results are derived from 100 independent Monte Carlo experiments.

### 4.1. Simulated Impulse Noise

In this experiment, we use impulse noise as the input reference signal **U**(*n*) to test the robustness of the proposed algorithm; according to [45], this noise can be understood as the superposition of two Gaussian noises with different variances. Therefore, we set the impulse distribution as **U**_1_ (*n*) ~ *h*(*n*)*N*(0,5) + *N*(0,0.1), **U**_2_ (*n*) ~ *h*(*n*)*N*(0,10) + *N*(0,0.01) and **U**_3_ (*n*) ~ *h*(*n*)*N*(0,15) + *N*(0,0.001), where *N*(μ, σ^2^) denotes a normal distribution with mean μ and variance σ^2^, and *h*(*n*) is a Bernoulli process with p_r_ as the probability that *h*(*n*) = 1, and we set p_r_ = 0.05 in this experiment. In addition, for the background noise **v**(*n*), we set **v**(*n*) ~ *N*(0,0.001).

Figure 3 demonstrates the effectiveness of the variant algorithm (H-FxAPGMC) when the input signal is **U**_1_ (*n*). In the early stages of iteration, all the compared algorithms achieved convergence (the simulation showed a trend of decreasing ANR). The proposed variant algorithm converged and reached a steady state after approximately 400 iterations. In comparison to the other benchmark algorithms, it reached a steady state more quickly and attained a lower steady-state ANR.

In the simulation comparison of the VSS-H-FxAPGMC algorithm, to demonstrate its robustness, we chose three different impulse noise runs in a non-stationary environment, as shown in Figure 4. The comparison results indicate that the proposed algorithm is least affected by the interference of pulse amplitude variations, as evidenced by its convergence speed and steady-state ANR, which is maintained around −60 dB. This is attributed to the combined enhancement of the hybrid and variable step-size methods designed by the modified MSD. Specifically, the variable step-size method presented in this paper considers the control of error weights, so this ensures that the algorithm is able to show better robustness and tracking performance when dealing with signals with large outliers. The experiment shows that the proposed algorithm satisfies the requirements of high performance and robustness at the same time, which is in line with the theoretical expectations of the experiment.

### 4.2. Real Industrial Noise

In this experiment, the superiority of the proposed algorithm is verified by using real industrial noise signals collected from a pile driver construction site at a building site. The noise is collected from the noise signal at the human ear in the office 50 m away from the construction site, and the sampling frequency of the microphone sensor is 44.1 kHz. One practical application of ANC technology is industrial noise reduction, and the effective control of this noise can improve the office environment and enhance work efficiency. The proposed algorithm has some significance in controlling industrial noise pollution.

Based on data from the microphone sensor, the actual noise sound pressure is displayed in Figure 5a. The noise waveform is both impulsive and periodic, with the impulse intensity approximately 30 times that of the background noise and a period of 1.3 s. The voltage signal was subsequently converted to sound pressure level, as illustrated in Figure 5b, and the sound pressure level of this noise was as high as 93 dB during the impulse period and as low as 65 dB during the inter-pulse interval. The spectral distribution of the noise is shown in Figure 5c, which is distributed in the frequency range of 0–8000 Hz, and the main component of the noise is 0–1500 Hz, and part of the noise corresponds to the ambient noise in the construction site, and the 1500–2000 Hz component can be regarded as the transition zone between impulse and ambient noise, and the component above 2000 Hz is the impulse noise generated by the operation of the pile driver.

In this experiment, considering the periodicity of this signal, we set the sampling rate to 4.4 kHz to accurately capture the noise characteristics and set the sample signal of 12,000 sampling points as the experimental data. Considering the complexity of the outdoor scene, we used **v**(*n*) ~ N(0, 0.001) as the background noise **v**(*n*) setting.

We evaluated the noise reduction performance of the proposed algorithm in the environment described above, and the results in Figure 6 compare the sound pressure of the sample signal with that of the residual signal after noise reduction. The amplitude of the residual signal is controlled within 0.1 when it reaches the steady state, which handles the impulse component well, but the interference of the impulse signal also causes the residual component of the impulse intervals to increase slightly in amplitude compared with the original sample signal. The result of this experiment is in line with the experimental expectations and can effectively eliminate the harsh impulse component of the site noise.

To further assess the noise reduction effectiveness of the proposed algorithm, we analyzed the frequency spectrum of the sample signal and the residual signal and made a comparison, as seen in Figure 7. From the comparison results, the amplitude of the residual signal is not obvious in all frequency bands, which means that the noise reduction performance achieves good results in all frequency band

Subsequently, we simulated the proposed algorithms under this noise, as demonstrated in Figure 8. The comparison algorithms include VSS-MFxAPSA, VSS-NFxAPSA, VSS-H-FxAP, and MVSS-H-FxAPSA. The simulation results show that the ANR of all comparison algorithms stabilize at a fluctuating state due to the impulse noise. In Figure 8, the convergence speed of all algorithms has reached an optimal state. Therefore, we proceed with the analysis and comparison of the steady-state ANR. As expected, the algorithm presented in this paper is the best in terms of processing performance, with a minimum ANR of −46.7 dB.

In addition, the residual noise waveforms of VSS-MFxAPSA, VSS-H-FxAPA, MVSS-H-FxAPA, and the proposed VSS-H-FxAPGMC are demonstrated in Figure 9. From Figure 9, it is evident that the noise control effect of the proposed VSS-H-FxAPGMC algorithm in this paper has the smallest residual component in the comparison. This result echoes the magnitude shown in Figure 8.

## 5. Discussion

This section presents the analysis and discussion of the experimental results. First, an analysis of the variant algorithm H-FxAPGMC is provided. The results in Figure 3 include a horizontal comparison between the H-FxAPGMC and APGMC algorithms, allowing for an intuitive observation of the performance improvement brought by the hybrid step-size decision in the variant algorithm. This improvement is significant: the H-FxAPGMC algorithm completes convergence in about 400 iterations, which is about 3500 points faster than the FxAPGMC algorithm, while ensuring a slightly better steady-state ANR than the FxAPGMC algorithm. This aligns with the second stage in Figure 2, validating the effectiveness of the hybrid step size and its performance gains. Compared to other algorithms, the H-FxAPGMC algorithm also demonstrates varying degrees of performance improvement, indicating its stronger denoising capability. Next, the effectiveness of the proposed VSS-H-FxAPGMC algorithm is assessed in Figure 4 using pulse noise with varying intensities. The increase in pulse intensity has a certain impact on the algorithm’s effectiveness. From the vertical comparison of the three results, it is evident that this effect is particularly noticeable in VSS-FxAPSA and VSS-NFxAPSA. However, the proposed algorithm maintains its convergence speed and steady-state ANR without significant performance degradation due to the increase in pulse intensity, and it remains in a leading position compared to the other algorithms.

Figure 8 compares the denoising performance of the proposed algorithm under real construction site noise conditions, which is summarized in Table 1. For the classical algorithms VSS-MFxAPSA and VSS-NFxAPSA, the performance of these two algorithms is slightly lagging behind; therefore, we focus on the differences between VSS-H-APA, MVSS-H-APA, and the algorithm proposed in this paper. The steady-state ANRs of these two advanced algorithms are from −32.4 dB to −42.5 dB and from −33.7 dB to −43.1 dB, with error values of 10.1 dB and 9.4 dB, respectively. The steady-state ANR of the VSS-H-FxAPGMC algorithm proposed in this paper fluctuates from −42.2 dB to −46.9 dB, with an error value of 4.7 dB. The error indicates the stability of the algorithm’s performance, demonstrating the superiority of the proposed algorithm in noise control. Compared with VSS-H-APA and MVSS-H-APA, the noise reduction performance is improved by about 19.2% and 16%, respectively. The intuitive comparison of residual noise in Figure 9 illustrates that the proposed variable step-size algorithm is more effective in handling pulse noise and outperforms existing algorithms.

In summary of the above experimental comparisons, the proposed algorithm demonstrates superior denoising capabilities across various input signals, including simulated pulse noise and real construction site sampling noise. This is primarily attributed to two key factors. First, the implementation of the variant algorithm is illustrated in Figure 3, where the improvement mainly impacts the early stages of the iteration. The hybrid scheme introduces a step size derived from MFxAPSA, which inherently possesses the property of suppressing outliers, thereby ensuring the robustness of the variant algorithm. The design concept of this scheme is based on step-size inequality analysis, enabling rapid convergence in the early iteration stages. Second, the variable step-size scheme, similar to most other variable step-size methods, adopts a modified MSD approach. The distinction lies in the introduction of a sliding control term to regulate the step-size equation. This approach makes the error term controllable within the iterative equation, which means the method is resilient to outlier interference, significantly reducing the performance loss caused by outliers.

In addition, we summarize the contributions of the existing literature and this paper in Table 2. The contributions of **the** existing literature include the design of a hybrid step size, the use of MSD minimization for step-size design, or the use of MSD correction for step-size design, while the contributions of this paper include both the hybrid step size and the introduction of a control term for designing the variable step-size method.

## 6. Conclusions

In the work of industrial noise control, the ANC system plays a great role, and the ANC algorithm, which is the core of this system, becomes one of the key designs. In this paper, the VSS-H-FxAPGMC algorithm is proposed by analyzing the APGMC algorithm, designing the hybrid step-size scheme, and analyzing and modifying the MSD. This paper provides a detailed analysis of the real-site noise data, examining its various aspects and assessing the residual sound pressure and residual spectrum following the implementation of the proposed ANC algorithm. In comparing the noise reduction performance of similar algorithms, the proposed algorithm has the least residual components of noise and the best noise reduction effect after realizing the noise reduction, which has an obvious effect improvement compared with other algorithms. During the field implementation, we consider the sensors’ arrangement in the house/building that will be disturbed by this noise for the specific implementation and application of the proposed ANC algorithm. In addition, we will also consider the combination with passive noise reduction to minimize the noise pollution. Regarding future work in this study, we will consider further optimizing the variable step-size scheme, continuing to explore improvements in the control term, and developing ANC algorithms with superior performance to better mitigate industrial noise pollution.

## Figures and Tables

**Figure 1 sensors-25-01881-f001:**
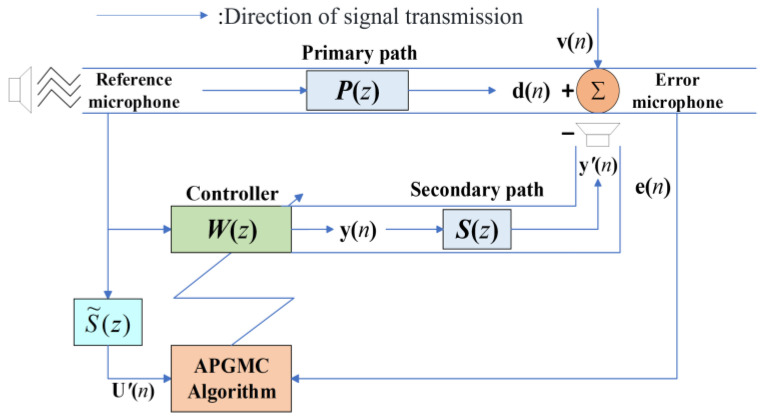
Schematic diagrams of ANC model.

**Figure 2 sensors-25-01881-f002:**
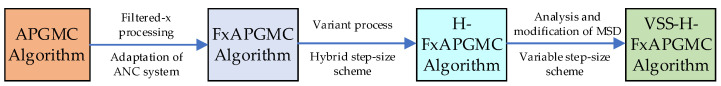
Summary of the overall algorithmic process.

**Figure 3 sensors-25-01881-f003:**
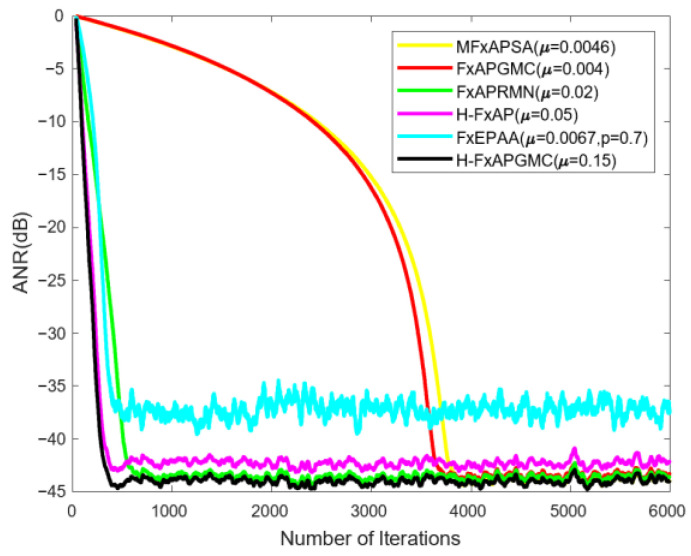
Performance of the variant algorithm with impulse inputs **U**_1_ (*n*).

**Figure 4 sensors-25-01881-f004:**
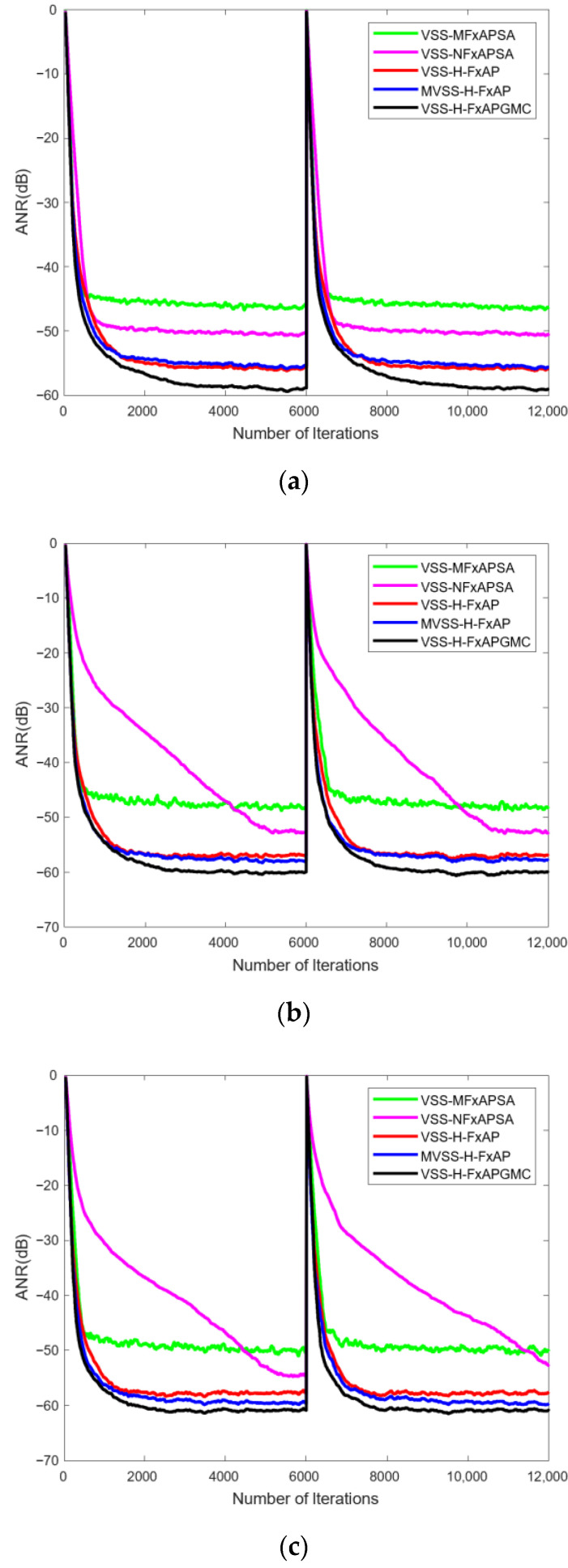
Performance of the VSS-H-FxAPGMC algorithm with three impulse noise inputs in a non-stable environment (from (**a**–**c**), input noise signals are **U**_1_ (*n*), **U**_2_ (*n*), and **U**_3_ (*n*)).

**Figure 5 sensors-25-01881-f005:**
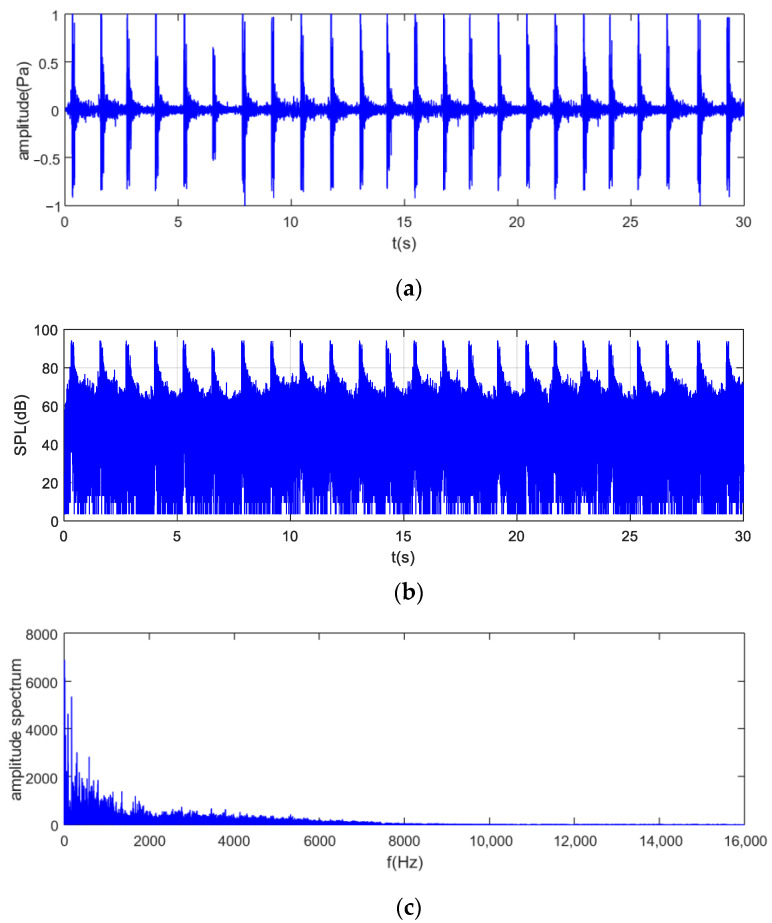
Real-site pile-driver noise data. (**a**) Sound pressure; (**b**) sound pressure level; (**c**) frequency–amplitude spectrum.

**Figure 6 sensors-25-01881-f006:**
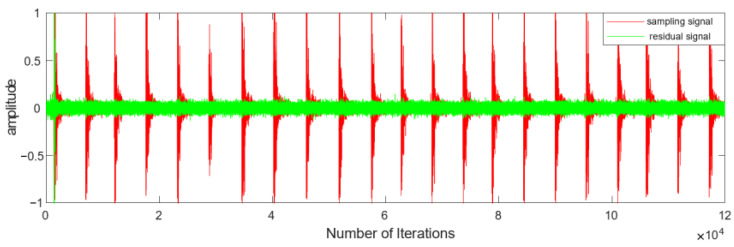
Sound pressure of the sample and residual signals.

**Figure 7 sensors-25-01881-f007:**
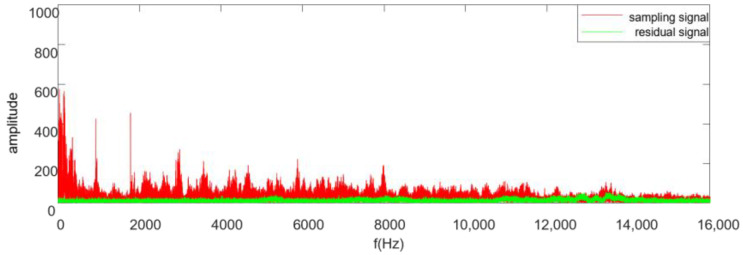
Frequency–amplitude spectrum of the sample and residual signals.

**Figure 8 sensors-25-01881-f008:**
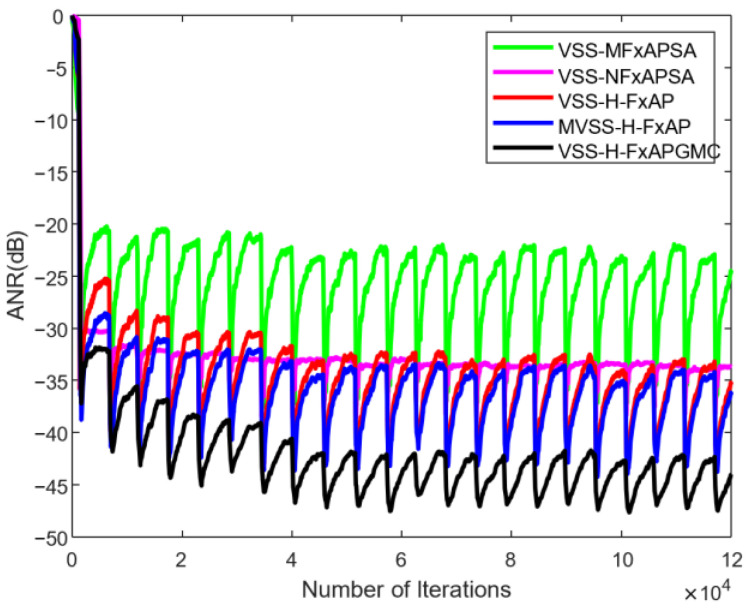
ANR of the proposed algorithm under real industrial noise.

**Figure 9 sensors-25-01881-f009:**
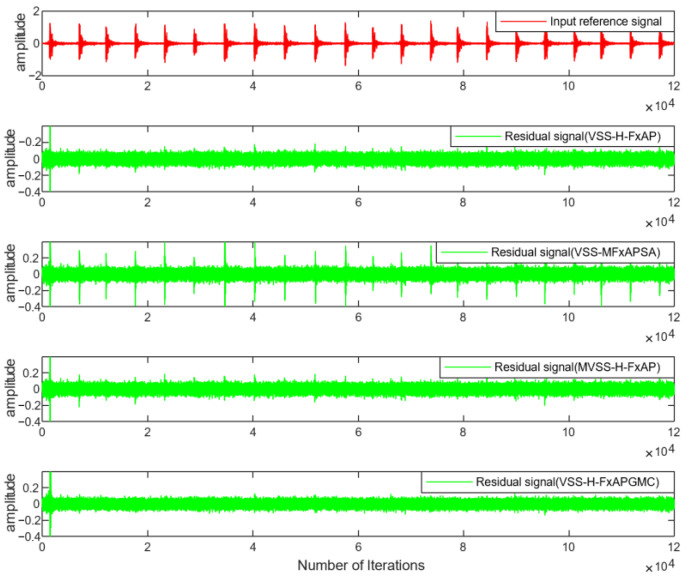
Waveforms of residual noise of different algorithms for real industrial noise.

**Table 1 sensors-25-01881-t001:** Comparative algorithm performance summary.

Performance Analysis of Each ANC Algorithm			
Algorithm	Maximum ANR	Minimum ANR	Error
VSS-MFxAPSA	−22.5 dB	−37.1 dB	14.6 dB
VSS-NFxAPSA	−32.9 dB	−37.0 dB	4.1 dB
VSS-H-APA	−32.4 dB	−42.5 dB	10.1 dB
MVSS-H-APA	−33.7 dB	−43.1 dB	9.4 dB
VSS-H-FxAPGMC	−42.2 dB	−46.9 dB	4.7 dB

**Table 2 sensors-25-01881-t002:** Summary of contributions to the literature.

The Contributions of the Various Literature in Addressing Pulse Noise	
Algorithm	Main Contributions
H-APA [22]	Proposed the application of hybrid step size in mixed-norm algorithm
VSS-H-APA [23]	Proposed variable step-size method for designing H-APA using MSD minimization
MVSS-H-APA [23]	Proposed variable step-size method for designing H-APA using modified MSD
VSS-H-FxAPGMC	Proposed hybrid step-size scheme for APGMC; proposed variable step-size method modified by a controlled error term

## Data Availability

Data are contained within the article.

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
