# Peer review of "Variable Step-Size Hybrid Filtered-x Affine Projection Generalized Correntropy Algorithm for Active Noise Control"

_sensors, 2025, doi:10.3390/s25061881_

Round 1
Reviewer 1 Report
Comments and Suggestions for Authors
The paper presents an active noise control algorithm for impulse noise, namely the VSS-H-FxAPGMC algorithm, which is of practical significance. The effectiveness of the method is verified by simulation. However, there are areas where the paper could be further improved, as follows:
1) The hybrid step-size can give the original algorithm a performance boost. Inspired by this idea, this paper combines the step-size of MFxAPSA with the FxAPGMC algorithm. Why choose the step-size of MFxAPSA instead of other algorithms? What are the advantages of the step-size of MFxAPSA? Please describe this briefly in Section 3.
2) P4: It is mentioned that "This variant scheme enhances the performance of the FxAPGMC algorithm, particularly in terms of computational complexity and convergence speed". Why this variant scheme can reduce the computational complexity? Compared with equation (8), equation (12) shows no change in computational complexity from the updating process of weight coefficients.
3) In Section 4, The comparison algorithms include two representative classes of algorithms: the classical ANC robust algorithms VSS-MFxAPSA, VSS-NFxAPSA, and the current state-of-the-art robust algorithms VSS-H-APA, MVSS-H-APA after filter-x in ANC. It is recommended that the author briefly give the basic principles of the comparison algorithm and make a brief introduction here rather than just listing the references.
4) Figure 2 shows the performance of the variant algorithm with impulse noise. Why are the values of the convergence coefficients of each algorithm in the figure different? How does the author choose the convergence coefficients of different algorithms?
5) It is suggested that when describing the simulation results, the authors should not only describe the control effect but also explain why the algorithm in this paper can achieve advantages. This needs to be analyzed in the simulation.
Reviewer 2 Report
Comments and Suggestions for Authors
1- This paper has a lot of definitions; it should have a paragraph to display the definitions.
2- Eq. 10 is not clear; how is this equation derived?
3- In the proposed section, the math model, especially equation 16, is not clear as well; the expectation of the \mu when the time is close to infinity is not correct; Is there any proof to manipulate eq.16 to eq. 17?
4- Moving from 20 to 21 to drive the final expression of \mu is confusing and needs more detail to show the derivation. Some variables are not defined; for example, in eq. 7, there is a \theta not specified in the below.
5- Some language errors are observed that should be considered for example abbreviations. In addition, the matching or similarity is too much during the authenticator software. It seems that the matching or similarity in proposed approach. This point needs to be clarified.
Comments on the Quality of English LanguageCan be improved.
Reviewer 3 Report
Comments and Suggestions for Authors
The works on active noise control as discussed need further testing at actual environment, although as shows in figure 4 the data from real industrial site but for complete and better results to check at least a round of testing required then compare the results to simulated.
Reviewer 4 Report
Comments and Suggestions for Authors
The authors present the article entitled “Variable Step-size Hybrid Filtered-x Affine Projection Generalized Correntropy Algorithm for Active Noise Control”. The article presents the following concerns:
-
The abstract of the manuscript does not present a clear methodology, the authors need to be clearer and more precise in the steps and methods to be followed, as well as if they take into account any international standards in relation to noise in construction, the acronym ANC is not described in the abstract, add. The authors do not highlight the novelty, nor do they show quantitative values of how much they improved with respect to other algorithms.
-
Please add quantitative values in the Abstract section in order to highlight the novelty of the work.
-
Authors are encouraged to describe the hypothesis and research problem they are solving in the paper. Also, please mention related works in the state of the art to highlight the novelty of the work. Finally, please mention the contributions of the work at the end of this section.
-
Although sections 2 and 3 describe the algorithms used, I consider it important to describe the general methodology of the experiment. Please add a flow chart explaining the general procedure and describing in detail the validation of the results.
-
Please add a discussion section where they provide an interpretation of the results. Also, in this section, I recommend adding a table that compares the main contributions of the work vs the already reported in literature in order to highlight the novelty.
-
Conclusion section: Please mention the future works related to this research.
-
Please reduce the similarity level to <20. In its current form, the manuscript presents 31% of duplicity.
Reviewer 5 Report
Comments and Suggestions for Authors
REVIEW
on article
Variable Step-size Hybrid Filtered-x Affine Projection Generalized
Correntropy Algorithm for Active Noise Control
Zhaoqing Mu, Ying Gao, Xinyu Guo and Shifeng Ou
SUMMARY
The article submitted for review is relevant to the problems of modern society and modern science. It presents a generalized variable step size correntropy algorithm with hybrid x-filtering and affine projection for active noise control.
The relevance of the problem is due to the noise from real construction sites. The scientific aspect of this study is due to the study of algorithms for efficient noise processing. The algorithm must have performance and reliability.
The authors set themselves the task of achieving algorithms that will outperform existing traditional algorithms. To do this, the authors introduced a hybrid step size into the generalized maximum correntropy affine projection algorithm. They conducted an in-depth study and demonstrated the real effectiveness of the proposed algorithm. A real construction site was tested and noise suppression control was performed. The results showed the effectiveness of the proposed algorithm.
The reviewer believes that the article is relevant, contains new scientific and practical data. However, this article has shortcomings that need to be corrected. The reviewer's comments are given below.
COMMENTS
1. The abstract is not well written. It does not formulate the scientific deficiency. The authors talk about the need to obtain a more advanced ANC algorithm, but they should show the existing scientific problems of the known algorithms. One sentence with the formulation of the scientific problem should be added to the beginning of the abstract.
2. At the end of the abstract, the authors talk about the effectiveness of the proposed algorithm for noise control, but do not show the percentage increase in this efficiency. The result should not be only listed but also given in quantitative characteristics.
3. The "Introduction" section is not detailed enough. There is no full analysis of the current state of the problem. The authors need to identify clearly the scientific deficiency of existing noise control methods. It is necessary to show the problem from the point of view of studying the processes occurring during noise control, and the scientific knowledge that is lacking in the modern scientific community. On the other hand, it is necessary to show the problem from an applied point of view for society and industry. This should be noted separately and given at the end of the "Introduction" section.
4. It is also recommended to add a review of 12-15 additional sources for the last 5 years to the Introduction section. It is necessary to show a more modern state of this issue.
5. The authors are strongly recommended to switch to the standard IMRAD research scheme. Now the article is difficult to perceive due to the fact that the methods and results are practically not separated from each other.
6. It is undesirable to provide an abbreviation in the title of paragraph 2. The authors need to think about how to change the name of the paragraph.
7. Paragraphs 2 and 3 look dry and are difficult to perceive. The authors need to add more analytics and describe the provided algorithms in more detail. The article should be attractive, including for readers from other fields of science. For example, for readers from the science of construction processes and construction organization.
8. The Experimental Results section contains many interesting graphs and dependencies, but they are poorly explained in the text.
9. I would like to see Figures 2 and 3 of higher quality. The same remark applies to Figure 6.
10. Figure 8 contains almost illegible small symbols; it should be provided in higher quality.
11. The absence of the "Discussion" section is noteworthy. This is unacceptable. A separate "Discussion" section should be provided and a detailed comparison of the obtained results with the results of other authors should be carried out. The main criterion for comparison will be the effectiveness of the proposed noise control algorithm. It would be desirable to provide a summary table or graph between the authors' results and similar studies.
12. The "Conclusions" section should be rewritten. It is necessary to formulate the scientific and applied results more clearly. Do the authors have any proposals for implementation at real facilities? This would strengthen the article.
13. The list of references, as already mentioned above, needs to be supplemented with additional sources for the last 5 years. The literature review should be more complete.
In general, the reviewer believes that this study has prospects, but there is still a lot of work to correct the comments. After correcting the comments, the reviewer would like to see the article again to make a final decision.
Round 2
Reviewer 2 Report
Comments and Suggestions for Authors
Still needs improvements and huge similarity.
Comments on the Quality of English LanguageLooks good.
Reviewer 3 Report
Comments and Suggestions for Authors
Accept to publish
Author Response
Thank you for ever asking a question and have a great day at work and in life!
Reviewer 4 Report
Comments and Suggestions for Authors
The authors addressed all my concerns. However, please consider the following observation:
-I consider that the authors have based some ideas from https://doi.org/10.1016/j.ymssp.2018.10.009. In lines 108-109, almost the same diagram is shown for ANC. The authors may be able to give credit to this work if they have used any ideas from it.
-Table 4: I suggest adding references in order to see the comparison.
